# Machine Learning in Resource-Scarce Embedded Systems, FPGAs, and End-Devices: A Survey

**Sérgio Branco** 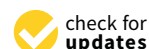**, André G. Ferreira \*** and **Jorge Cabral**

Algoritmi Center, University of Minho, 4800-058 Guimarães, Portugal; asergio.branco@gmail.com (S.B.); jcabral@dei.uminho.pt (J.C.)
**\*** Correspondence: id4541@alunos.uminho.pt; Tel.: +351-915-478-637

**Abstract:** The number of devices connected to the Internet is increasing, exchanging large amounts of data, and turning the Internet into the 21st-century silk road for data. This road has taken machine learning to new areas of applications. However, machine learning models are not yet seen as complex systems that must run in powerful computers (i.e., Cloud). As technology, techniques, and algorithms advance, these models are implemented into more computational constrained devices. The following paper presents a study about the optimizations, algorithms, and platforms used to implement such models into the network's end, where highly resource-scarce microcontroller units (MCUs) are found. The paper aims to provide guidelines, taxonomies, concepts, and future directions to help decentralize the network's intelligence.

**Keywords:** machine learning; embedded systems; resource-scarce MCUs; FPGA; end-devices

## 1. Introduction

The digital revolution is becoming more evident every single day. Everything is going digital and becoming internet-connected [1,2]. In the year 2009, the number of devices connected to the Internet has overpassed the world population; therefore, the Internet no longer belonged any specific person, and the Internet of Things (IoT) was born [1]. The number of internet-connected devices keeps rising, and by 2021 the number of devices will overpass 20 billion [3].

Because of all these devices, internet traffic will reach 20.6 Zettabytes. However, IoT applications will be generating 804 ZB. Most of the data generated will be useless or will not be saved or stored (e.g., only 7.6 ZB from the 804 ZB will be stored) [3]. Moreover, only a small fraction will be kept in databases. However, even this fraction represents more than the data stored by Google, Amazon, and Facebook [4]. Because humans are not suited to the challenge of monitoring such a high amount of data, the solution was to make machine-to-machine (M2M) interactions. These interactions, however, needed to be smart, to have high levels of independence.

To achieve the goal of providing machines with intelligence and independence, the field of artificial intelligence (AI) was the one to look for answers. Inside the AI field, the sub-fields of machine learning (ML) and deep learning (DL) are currently the ones where more developments have happened in the last decades. Besides the fact that the topic of AI has gained traction only a few years ago, machine learning algorithms date back to 1950 [5]. However, the AI field was immersed in a long winter because computers were not powerful enough to run most of the ML algorithms [6].

Once the computers started to have enough memory, powerful control processing units (CPUs), and graphical processing units (GPUs) [7], the AI revolution started. The computers were able to shorten the time to create and run ML models and use these algorithms to solve complex problems. Furthermore, cloud computing [8] allowed the overcoming of the challenges of plasticity, scalability, and elasticity [9,10], which has encouraged the creation of more flexible models that provide all the

resources needed. The Cloud provides limitless resources by the use of clusters, a network of computers capable of dividing the computational dependencies between them, making these computers work as one [11]. This technique made possible the creation of large databases, the handling of large datasets, and to hold all the auxiliary calculations made during the model building and inference phases.

However, to build a good ML model is not the computer power that matters most; but the data used to build it [12]. The Cloud had shortened the building and inference times and provided the memory necessary to store the data, but not to create the data. The data source is the IoT devices, which are collecting data, 24/7, around the globe. The full network architecture is depicted in Figure 1.

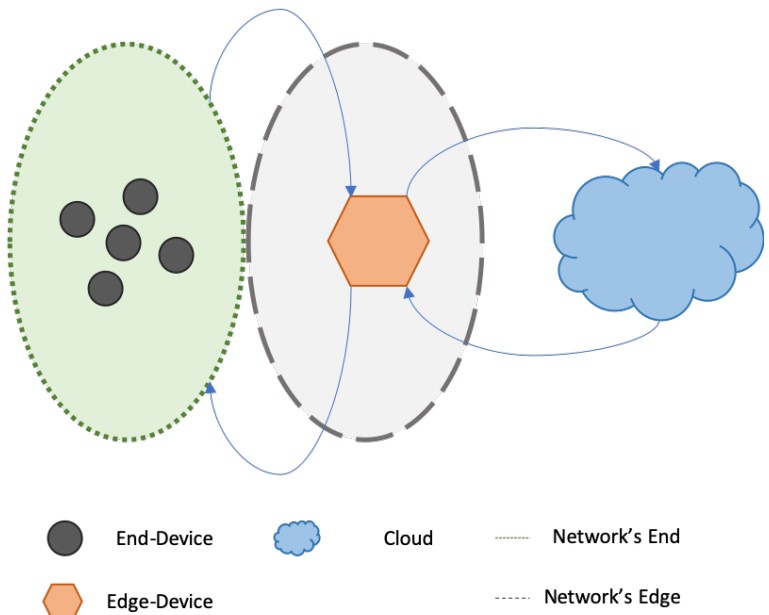

**Figure 1.** Standard Internet of Things (IoT) network architecture depiction. The network has many boundaries, the Cloud, the edge, and the environment/end. The end-devices are responsible for acquiring the environmental data; send the data to the edge-device, which will be forward the data to the Cloud. The Cloud can then send a response to the edge and the end-device. Other architectures can be set, but, this is the architecture used in the scope of this document.

The first approach, to create intelligent networks, was having end-devices collecting data through the use of sensors. Because these devices are resource-scarce, they cannot store the data; they will broadcast the data to the edge-device. The edge-device will then forward the data to the Cloud via an internet connection. The Cloud will receive the data and process it, either for storage or to feed to an ML model. The ML model will then run the inference on the data, and achieve an output. The output could then be broadcast back to the end-device, or sent to another system/user; this means that the system would be entirely cloud-dependent.

The major problem with a cloud-dependent system is that it relies on an internet connection for the data transfer [13]. Once this connection gets interrupted, delayed, or halted, the system will not function properly. The internet communication field is continuously receiving new developments to improve internet connection stability, traffic, data rate, and latency. Besides these developments, we all have been in situations where, even with a strong internet signal, we do not have a reliable connection.

A system architecture, in which its intelligence is cloud-based, inherits all the problems imposed by an internet connection. Besides the time the request takes to reach the Cloud and the response to be received, there are also privacy and security concerns [14,15]. For the Cloud to be accessible, it must have a Static and Public IP address [16,17]. Therefore, any device with an internet connection can reach it.

However, nothing can travel faster than light, even in an ideal environment a byte would take at least *2\*distance/light_speed* to reach the Cloud, and to be sent back to the end-device. The further the

end-device is from the Cloud, the higher the communication's time overhead. Computer scientists have encountered a similar problem. The processing power of CPUs was rising, but that did not matter because of the time it took to access the data [18–20]. One of the solutions found was to cache the data and to create multiple cache levels, each one smaller than the other, as they got closer to the CPU [18–22].

Cloud computing (Figure 2) provides advantages that cannot be easily mimic, which should be used to improve any application. However, in real-time/critical applications, the Cloud may not be the best place to have our entire system's intelligence, because we lose time, reliability, and stability. So, like the cache, we may have to store our models closer to the end-device.

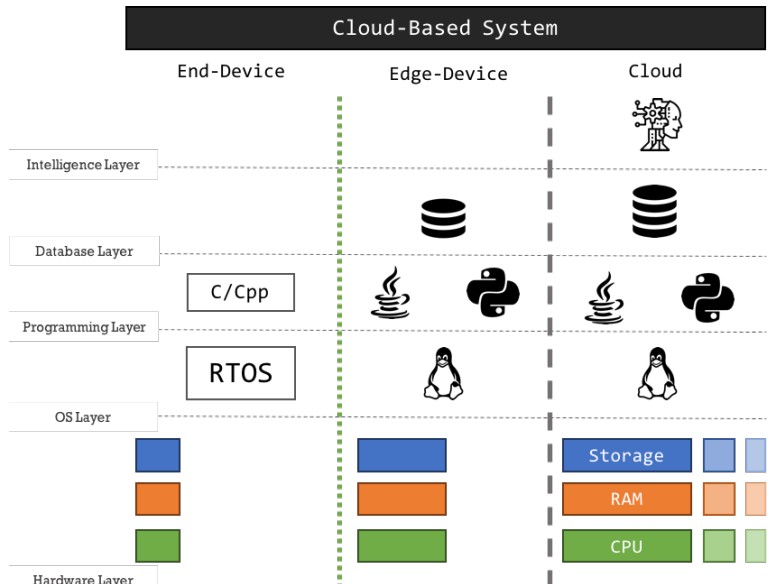

**Figure 2.** Cloud-based system depiction. In a Cloud-based system, the intelligence is entirely in the Cloud, where *limitless* RAM, memory storage, and CPU exist. The Cloud is responsible for data storage and processing. The high-level languages (e.g., Python or Java) are used to shorten the development time and to improve the system's security. The edge-device is similar to the Cloud, however, its hardware resources are finite.

## 1.1. Fog and Edge Computing

Edge-devices can play distinct roles inside a network. Usually, they are the network's gateway. Therefore, they are responsible for receiving the data from the end-devices and forwarding the data to the Cloud. However, edge-devices may be responsible for temporarily storing the data (in small databases for local usage), data preprocessing, network security, set up, and ensuring the well-being of the network. Because of the many roles and processing needs, edge-devices can range from personal computers to microcomputers (e.g., Raspberry Pi). They can handle multi-threading applications, have fast CPUs, and large memory storage capacities. Therefore, we cannot consider these devices to be resource-scarce. The edge-devices run standard operating systems (OS). A Standard OS allows multiple programs to run *simultaneously*, to have access to more software-resources, and to be programmed with high-level languages, such as Python and Java. These are the same languages used to program the Cloud system. Therefore, exporting an already built ML model, from the Cloud to these devices, is pretty simple.

As stated in Section 1, a Cloud-based system has many drawbacks, especially in terms of internet connection latency, data privacy and security, downtime, and attacks (i.e., man-in-the-middle and DDoS attacks). Because of these problems, inherit from Cloud-based systems, the simplicity to create the models in the Cloud and deploy them in the edge-device, developers have started to bring the intelligence to this network layer.

Edge-computing reduces communication latency by dividing the ML model into multiple layers. The bottom layers are implemented in the edge-device and the top layers in the Cloud. Other approaches go further, and the bottom layer is implemented in the end-devices; this method is called fog-computing (Figure 3) [23–26]. Therefore, the system's security [27] is improved because the data does not have to leave the local network. If the data has to leave the local network, the privacy can be increased with the use of deep neural networks' (DNN) semantics [28]. Because of these, a DDN uses a technique called feature extraction, and only the DNN can understand the data. Therefore, by implementing the bottom DNN layers in the edge-device, and passing the features extracted to the Cloud, even if those features are stolen, no one can take anything from them.

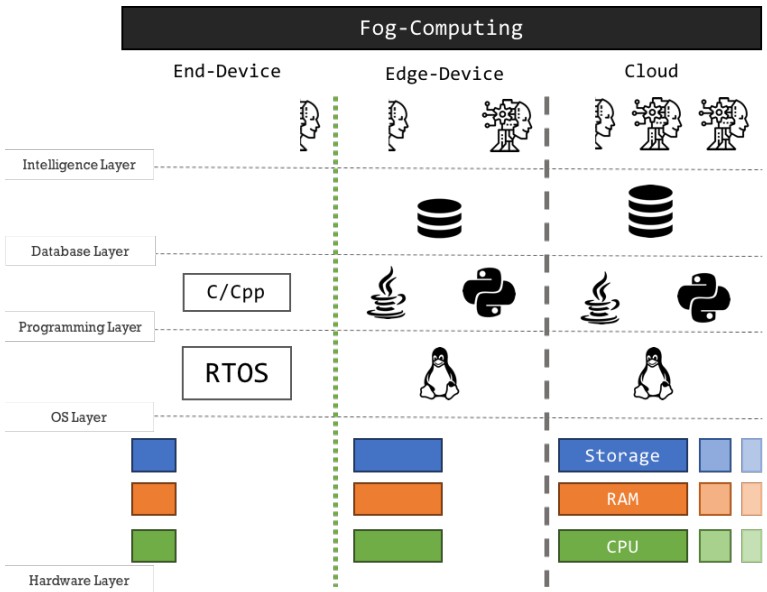

**Figure 3.** Fog-computing depiction. Contrary to the Cloud-based system (Figure 2), in fog-computing, the machine learning (ML) model is divided into layers, and the bottom layers can be implemented in the edge and end-device. However, they are not stand-alone ML models, and the inference is still Cloud dependent.

Fog-computing has gained traction with Industry 4.0. This new revolution in the factories has come with the need for security, privacy, time, and intelligence; cloud-based systems could not fulfill all these needs. Fog-computing helps factories reduce their waste, economic losses, and improve their efficiency [29]. However, these edge-devices have no worries in terms of power-consumption efficiency (they are plugged into the power grid). Furthermore, the call-to-action is done by employees inside the factory, whenever the system finds a problem that needs a solution.

*1.2. End-Devices*

In most applications, bellow an edge-device exists an/several end-devices, which collect the application-specific data. The interaction and monitoring of the physical world are made through actuators and sensors, who use a broad set of communication protocols to send their readings or to receive orders.

The end-devices are usually battery-powered because they stay in remote locations. Therefore, these devices must have low power-consumption, deep-sleep, and fast-awakening states. They should spend energy executing only the task for which they were designed; this means they must be application focus. The end-device has to accomplish the deadlines, to achieve the goal of a real-time application. Therefore, they either run a bare-metal application or a real-time OS (RTOS) [30], instead of a standard OS. The RTOS allows them to multi-task, prioritize the tasks, and to achieve the deadlines. A standard OS (e.g., Linux) cannot ensure the task's deadline is met, or that the highest-importance task

has the entire CPU for themselves when running [31,32]. Furthermore, there are other limitations set to these devices, such as size, cost, weight, maximum operation temperature, materials, and pin layout.

To fulfill the goals mentioned above, the end-device's microcontroller unit (MCU) is generally small and resource-scarce. The MCU has small memory storage and RAMs, low-processing power, but a broad range of standard input/output pins to fit any application needs. Resource-scarce MCUs fit the application's constraints, but they also impose limitations in the development and software. Like in many computer-science fields, the tradeoff is the term that guides the system.

### 1.3. Why Machine Learning in the End-Device?

The Cloud provides limitless computational resources. The fog/edge computing can reduce latency, improve security and privacy, and the exportation of cloud-generated ML models to the edge-device is easy. Why should we bother to implement ML stand-alone models into resource-scarce MCUs?

The Cloud/edge-based intelligence systems do not ensure the achievement of real-time deadlines; this is a primary issue for real-time applications. The time-reliability, in these systems, is influenced by many variables; they are unpredictable and random. Therefore, the mitigation of their influence is hard or impossible. Furthermore, for the response to reach the edge-device and the end-device to receive the response, a communication channel is needed. This channel usually relies on wireless communication technology. Studies have shown that 70% of the end-device power consumption in the wireless communication network is spent on communication [33–35]. Therefore, in systems where the communication channel needs to be wireless, and the end-device is battery powered, reducing the number of requests/responses is important, because it is cheaper to process the data than send it.

The implementation of intelligence in the end-devices/resource-scarce MCUs is the solution to mitigate these problems, and many others discussed in Section 1. Furthermore, some applications (e.g., medical) have small deadlines and do not allow the use of wireless communications.

### 1.4. Scope

There is literature on diverse topics about ML in IoT Networks. A recent survey, [27] describes the tools and projects where the intelligence was brought to the network's edge. The paper focuses solemnly on edge-computing (with brief discussions on fog-computing). The paper shows the importance of bringing ML models to the network's edge, and some of the key concepts in this paper can cross-over to the main topic of this one. Because some of the tools described in [27] paper can cross-over the edge and be used in resource-scarce MCUs (section), we included the ones available for both platforms in this paper and provided the showcases where those tools were used.

Other surveys [36–42] provide the fundamentals for machine learning, deep learning, and data analysis in IoT Networks, but the focus is generally on Cloud-based systems, or fog-computing. The surveys do not focus on the end-device, especially on the optimization techniques, challenges, and importance of bringing ML models to the end-device. However, these surveys present much information on applications, and topics such as security, robustness, and independence on such systems. Therefore, they are essential to the field, and some ideas and concepts are used and transposed for this paper.

This paper aims to provide insights on how to implement stand-alone ML models into resource-scarce MCUs and field-programmable gate arrays (FPGAs) that are typically at the end of any network. The document presents tools available to export the models from the Cloud to the end-device's MCU. The approaches a developer can take to achieve the goal. How the ML models can be helpful to mitigate some problems any network faces.

The authors decided to focus on the end-device because the literature about the topic is scarce, and they believe that more focus should be given to this research area; because of the topic's importance to the development of more robust and intelligent systems. Moreover, the creation of intelligent

end-devices can be the best way to have real-time applications and low-power consumption in IoT Networks, and use the true potential of this technology.

## 2. Machine Learning

The literature about machine learning algorithms, their origin and history, and their influence on several fields is extensive [12,43–45], and out of this paper's scope. However, the following sections use terms and concepts that not everyone is aware of, outside the AI field; for this reason, the authors provide this section, which briefly explains some concepts, algorithms, and terms used in the field.

### 2.1. Concepts

The first concept to have in mind when discussing the ML theme is that ML is not a single algorithm. To build an ML model, we must choose from a set of algorithms the one to use. Moreover, besides the multiple projects and areas where they were applied to already, we cannot make any assumption on how the ML algorithm will perform. The no free lunch theorem [46,47], as it is known, states that: Even if we know a particular algorithm has performed well in a field, until we create and test a model, we cannot state it will perform well in our problem. From all the algorithms, the most commonly used/known are support-vector machines (SVM), nearest neighbors, artificial neural networks (ANN), naive Bayes, and decision trees.

Another key concept is: data is the backbone of any ML model. The model can only perform as good as the quality of the data we fed to it during the learning phase (more about this phase can be found in Section 2.2). Furthermore, any ML algorithm will achieve the same accuracy as the others, if the algorithm is fed with enough data [48,49]. Therefore, since the quality of the data is crucial for the algorithm performance, most of the model development effort is towards the data extraction and cleaning. Building and optimizing a model is just a small part of the whole process.

Machine learning's development is divided into two main blocks. The first block is the model building, where data is fed into the ML algorithm, and a model is built from the data. The second block is the inference, where new data is given to the model, and the model provides an output. These two blocks are not strictly independent, however, they are not dependent either. At the conclusion of the model building block, the algorithm provides a representation of the data. Afterward, the inference block uses the data representation to achieve the outputs. In the great majority of ML algorithms, the inference results usually are not used to rebuild or improve the model, but in some algorithms, they are. Therefore, the inference block can only start after the model is built, and the inference's results can rebuild the model, but there is a separation between the two blocks, which allow them able to run on different machines.

### 2.2. Model Building

Building a machine learning model may take a few steps, but we are only going to focus on the two most important ones: training/learning and validation/simulation phases. The learning phase is the most critical because it is the phase responsible for generating the data representation. As discussed earlier, to obtain a powerful data representation, the data used during the learning phase must have a high-quality. In the learning phase, the computational resources necessary to achieve the goal are high. The number of calculations is massive; the memory consumption is large (increases with the data complexity); the processing power and time to finish the learning phase have inversely proportional curves. The validation phase is the one who shows the accuracy achieved by the model. During the building block of an ML model, the primary dataset is split into two (usually 75/25), where 75% of the data is used in the Learning Phase, and 25% in the validation phase.

#### 2.2.1. Learning Phase

An ML algorithm can be classified according to its **learning nature**. The learning nature describes if the algorithm needs a primary dataset and if the dataset must have the expected output included.

If the algorithm's nature is supervised (*supervised learning*) [43,50,51], the algorithm needs a primary dataset, and all the *instances* must have the corresponding output. If the algorithm does not require the primary dataset to have the outputs, and the algorithm intends to cluster the data into subsets, we say the algorithm has *unsupervised learning* [43,50,51]. However, sometimes having someone labeling the data can be time-consuming and costly. To reduce the cost and time dispensed labeling data, it is possible to label just part of the dataset, and then let the algorithm cluster the remaining data into each class. This method is called *semi-supervised learning* [43,50,51].

Moreover, the algorithm may not need a primary dataset; the learning is incremental, and the machine learns through trial-and-error while collecting the data. Every time the machine does something wrong, it gets a penalty; every time the machine does something right, it gets a reward. The term used to describe this technique is *reinforcement learning* [52].

Distinct algorithms have distinct **learning-dynamics**. The algorithm's learning-dynamics provides insights about its plasticity, and how it handles the primary dataset. If the algorithm learns in a single batch, to improve or reconstruct the model, the developer must restart from scratch. Therefore, once the model is built, the algorithm cannot relearn anything new; this is called *batch learning* [51]. If the algorithm can learn on-the-fly; therefore, it can be built and rebuilt as it obtains new data, it is called **online learning** [51].

The data representation generated after the learning phase can have different natures. Therefore, a model can have two main **data representation natures**. If the model stores the primary dataset, and compares the new data with the previous one during inference, we consider our model to be *instance-based*. However, if the model infers a function two obtain the output from the data features, that is called *model-based*. In some situations, the algorithm can create a mixed of instance and model-based data representation. The data representation nature allows the developer to know how much memory a model uses.

The **tribe nature** [45] of an ML model describes how the model gets its knowledge, therefore, this can tell the developer the architecture the model has after the learning phase. If the model is a *symbolist*, the inference is made through a set of rules; therefore, the model is similar to multiple if-then-else programs. An *analogizer* model picks the new data and projects it against a data representation, and compares the new data with the previous one. Therefore, in its structure, the model stores a representation of the training dataset. A *connectionist* model uses a graph-like architecture, where dots connect in multiple layers, from the bottom (input) to the top (output); this layer structure allows the splitting of the model. The *Bayesian* model calculates the probability for the input to represent each class, because of the probabilistic inference these models do, they use in their cores mass calculations. The last tribe is the *evolutionaries*; these models can change cross chunks of code (genes) to obtain new programs to solve the problem they have in hand. Therefore, their final structure is very modular.

The Figure 4 compiles the natures a ML algorithm can have, summarizing the text above.

2.2.2. Validation Phase

The model's validation phase is what allows the developer to know how good the algorithm has performed on the data. Until the end of this phase, we cannot make any assumption on the algorithm's performance. The standard metric used to describe the algorithm's performance is the overall accuracy. This metric is the percentage of rights classifications over misclassifications. However, the use of other metrics can provide better insights into the model's overall performance. Some of those metrics are: confusion matrix, that shows which classes/instances our model has the worst performance; area under ROC curve to test the validity of binary classification models; mean absolute/square error are the ideals to test the model's performance, in regression problems [53–55].

### 2.3. Programming Languages and Frameworks

The set of programming languages and frameworks for ML development is vast and is increasing day-by-day [56]. Today, it is possible to create any ML model with almost no background in the AI field or mathematics. Furthermore, these tools are available for many programming languages. In this section, we discuss the most used tools in the field of ML to achieve the goal of model building.

Python is one of the most used languages to create ML models. Therefore, multiple packages and frameworks were developed for this language. Scikit-Learn [57] is a package which contains all the basic algorithms used in ML applications. For more complex applications, PyTorch [58] and TensorFlow [59] provide ways to simplify deep learning, natural language processing, and computer vision. TensorFlow also has implementations for C++ and Javascript. Caffe [60] is a framework for deep learning, the framework was written in C++, but its interface is for Python.

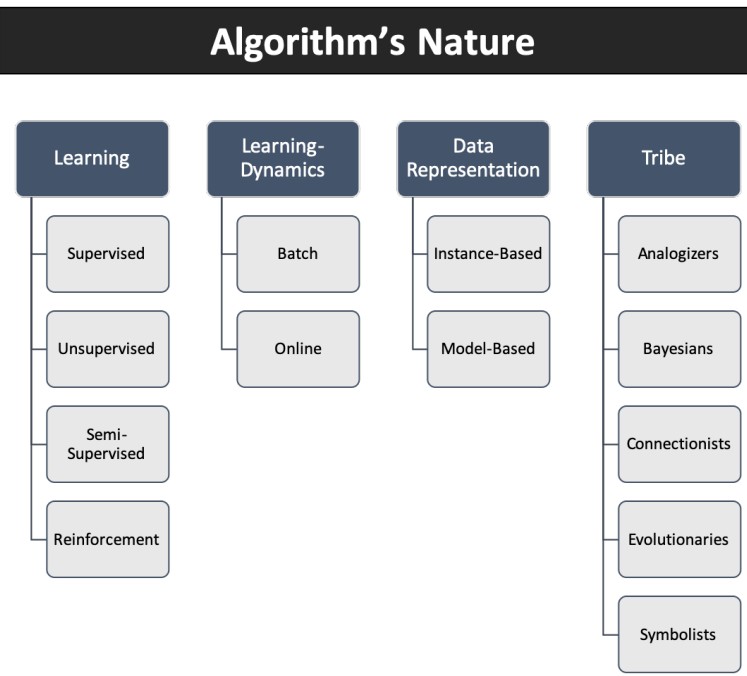

**Figure 4.** Algorithm's natures.

Besides Python, R [61] is a programming language widely used for statistical computing; therefore, many data analysts and statisticians use R to perform data analysis and build statistical models. Moreover, R allows manipulating its objects by using other languages, such as C/C++, Python, and Java.

Julia [62] has gained traction among machine learning, statisticians, and image processing developers. This programming language has high-performance, allows the call of Python functions through the use of PyCall. Furthermore, Julia Computing has won the Wilkinson Prize for Numerical Software because of its contributions to the scientific community in multiple areas, among them deep learning.

The Java programming language has Weka [63], a powerful tool for machine learning development and data mining; Knime [64] is also commonly used by Java developers to bring intelligence to their system. For C++, Shogun [65] is the most used library because it implements all ML algorithms. However, some libraries are available for each algorithm. One of the most known is libsvm [66], for SVM model development.

*2.4. Exporters and Transpilers*

Section 2.3 has shown which languages, libraries, and frameworks are used to build ML models. Python dominates the field of ML model building, but several options are available for other languages. The use of high-level languages, (i.e., Python, Julia, R, and Java) is due to the simplicity of handling the data in these languages. Since model building relies on data treatment, database queries, and data visualization, those languages have other tools to accomplish those tasks. In C/C++ (low-level languages), the same is not done quickly, which increases the development time.

However, C/C++ are the languages used to program most resource-scarce MCUs and FPGAs, because these languages allow access to the hardware and manipulate it directly. Moreover, these languages have less overhead than others, and their performance tends to be better. Therefore, there is a discrepancy between the programming languages used to build ML models, and the ones used to program the end-devices.

Moreover, as stated previously, the ML model building is done in two distinct phases (learning and validation). The learning phase is the one more computational dependent because of all the data, and calculations made to understand the data behavior and generate the data representation used during inference. Therefore, build a model directly in scarce-resource MCUs would be difficult and would take much time to accomplish. A solution to this would be to build the model in an high-computational space (such as cloud computing) and then implement the inference code and data representation in the end-device's MCU. Because of the use of high-level languages in the Cloud, and low-level languages in the end-devices, there is the need for transpilers/exporters [67], to convert high-level code/objects to low-level.

It is essential to notice that transpiling code and transpiling objects are distinct (Figure 5). Code transpiling means that the source code from language A is converted to source code to Language B. For example, a print ("Hello World"), in Python 3.x, turns to a printf ("Hello World") in C. However, transpiling an object means that the objects code and data, generated after the program execution, are both converted, and not only the generation code. Object transpilers are the ones we want to use in our case.

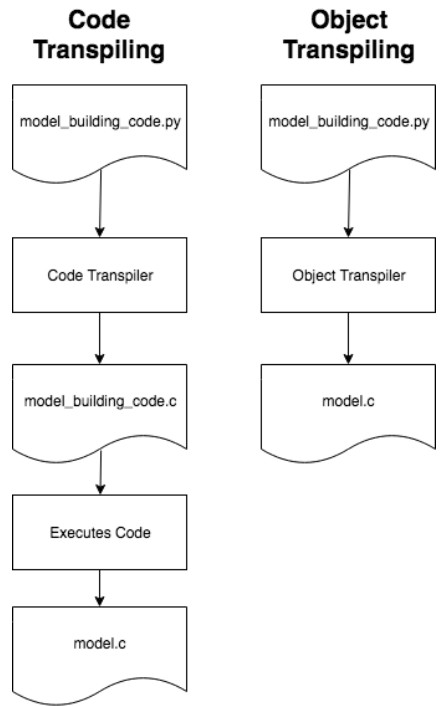

**Figure 5.** Transpiler's workflow. The workflow of both transpilers, as it is possible to see, transpiling objects means that the model build part is done directly in the original programming language, and only the data object is transpiled.

There are some transpilers available. Sklearn-porter [68] is a package for Python, which allows to transpile models trained with Scikit-Learn to other programming languages, among them are C/C++. Weka-Porter only allows to convert Decision Trees models to C/C++, therefore as many limitations. MATLAB Coder is an exporter for MATLAB to C; however, it is not assured to be able to export trained models directly to C.

There are some options to transpile trained models from high-level languages to C. The transpilers have many limitations and do not make any code optimization, they are usually generic, and try to create code to run in any platform. Therefore, any code optimization must be done by the developer.

## 3. Machine Learning in End-Device Embedded Systems

Besides the nomenclature resource-scarce (i.e., resource-constraint) MCUs/end-devices is used many times in literature, there are disparities on what can be a resource-scarce MCU/end-device. Therefore, a correct definition is needed. Some papers define that the IoT networks' end-devices are usually resource-constrained, with memories in the orders of Kilobytes (KB), low-clock frequencies, and low-power consumption [69–73].

However, some papers [74–78] refer to resource-scarce (or similar nomenclature), but they use devices such as Raspberry Pi (RPi), which cannot be considered as resource-scarce in the scope of this paper, and in many IoT applications. Moreover, an RPi or a BeagleBone can run high-level languages like python, which simplifies the process of training the model in a Cloud computer, and export it to these platforms through the use of tools like Pickle [79].

In Table 1, a list of MCUs and development boards associated with papers describing ML implementations in resource-scarce devices is shown. As we can see, the capabilities of the different MCUs and development boards varies considerably (e.g., RPi has considerably more resources than the other MCUs or development boards). Therefore, in the scope of this paper, the maximum specifications of a resource-scarce device are: a clock frequency under 200 MHz; on-board flash <3 MB; power consumption under 20 mA (or that had a deep-sleep/low-power state); (S)RAM memory no larger than 500 KB. At the light of the current state in technology, these were the metrics chosen to describe a resource-scarce MCU. However, because technology keeps evolving, maybe in the future, the metrics will be the MCUs size and power-consumption.

**Table 1.** Microcontroller units (MCUs) comparison. The table was ordered by the MCU's clock frequency.

| Name | Clock Frequency | Flash | (S)RAM | Power |
|---|---|---|---|---|
| Arduino Uno (ATMega128P) | 16 MHz | 32 KB | 2 KB | 12 mA [80] |
| Arduino Mega (ATMega2560) | 16 MHz | 256 KB | 8 KB | 6 mA [81] |
| Arduino Nano (ATMega2560) | 16 MHz | 26–32 KB | 1–2 KB | 6 mA [81] |
| STM32L0 (Cortex-M0) | 32 MHz | 192 KB | 20 KB | 7 mA [82] |
| Arduino MKR1000 (Cortex-M0) | 48 MHz | 256 KB | 32 KB | 4 mA [83] |
| Arduino Due (Cortex-M3) | 84 MHz | 512 KB | 96 KB | 50 mA [84] |
| STM32F2 (Cortex-M3) | 120 MHz | 1 MB | 128 KB | 21 mA [85] |
| STM32F4 (Cortex-M4) | 180 MHz | 2 MB | 384 KB | 50 mA [86] |
| RPi A+ | 700 MHz | SD Card | 256 MB | 80 mA [87] |
| RPi Zero | 1 GHz | SD Card | 512 MB | 80 mA [87] |
| RPi 3B | 1.2 GHz | SD Card | 1 GB | 260 mA [87] |

The section below describes projects, optimization techniques, and solutions of stand-alone ML models deployed in resource-scarce MCUs and FPGAs. The following section refers to both platforms as embedded systems (besides the term represent a vaster majority of microcontrollers). The section also presents novel ML algorithms (e.g., algorithms explicitly created for embedded systems). The authors intend to give the reader information on the projects' development, the

algorithms implemented, the time required to output a prediction, the accuracy, and how the optimizations achieved and why were those optimizations necessary.

In the Appendix A, the authors have compiled a table (Table A1) which shows the projects, optimizations, accuracy, and performance obtained for each project. Because of some papers present more than one optimization or algorithm implementation, the authors have decided to divide the table according to each achievement.

### 3.1. Technical Review

The past few years have brought new resource-scarce MCUs that had focused on improving their ML capability and compatibility. This section provides a few insights about some of the MCUs available for ML development and shows how the field is evolving.

In 2017, Microchip has implemented the first MCU with a high-performance 2D GPU, the *PIC32MZ DA* family [88]. Because GPUs can handle parallel calculations, the use of GPU technology in ML is important and can improve the system's performance in terms of execution time. Therefore, by implementing low-power tiny MCUs with GPU capabilities in them, can help give a step forward to implement more complex models in the end-devices.

ARM has recently (2019) launched its Helium technology. This technology is intended to be present in the next generation of ARMv8.1 MCUs (Cortex-M), providing higher digital signal processing and machine learning capabilities to their MCUs. This technology can achieve up to $15\times$ better performance in ML applications, and up to $5\times$ in DSP applications [89].

ETA Compute has released its ASIC + ARM Cortex-M3 ECMxx MCU models that can be applied in applications such as sensor fusion, speech recognition, healthcare, and video. They claim their products are low-power, which makes them usable in battery-powered systems [90]. The use of ASIC technology for domain-specific applications is shown to be helpful as it allows the ability to reduce power-consumption, size, and improve performance.

Currently, the market offers sensors with incorporated ML cores in them, for continuous monitoring. STM has launched (2019) two sensors (3D Accelerometer + 3D Gyroscope), with an embedded decision tree classifier. Decision trees are usually connoted with if-then-else statements, making them easy to implement, but powerful models. These sensors can be configured to run up to eight decision trees. These DTs can have a maximum number of 512 nodes for all the DTs implemented/configured [91,92].

The advances in the implementation of modules, for the ML domain in resource-scarce MCUs, is still recent. The market is starting to respond to this new trend. The authors believe that, shortly, more devices and technology for the implementation of ML models in resource-scarce MCUs will appear.

### 3.2. Novel Machine Learning Algorithms and Tools for Embedded Systems

To understand the optimizations and the prosperity of ML in scarce-resource embedded systems; we must take a look into the libraries/tools already provided to accomplish this task. This section presents tools and libraries that are gaining traction, and that are mentioned in the following sections. The authors have focused on the main optimizations made; and not on the foundation for these novel algorithms and libraries implementation.

ProtoNN [93] is a novel algorithm to replace kNN (nature: supervised/unsupervised; batch; instance-based; analogizer) in resource-scarce MCUs. This algorithm stores the entire training dataset, and use it during the inference phase; this technique would not be feasible in memory space-limited MCUs. Because kNN calculates the distance between the new data point to the previous ones; ProtoNN has achieved optimization by techniques such as space low-d projection, prototypes, and joint optimization. Therefore, during optimization, the model is constructed to fit the maximum size, instead of being pruned after construction.

Bonsai [94] is a novel algorithm based on decision trees (nature: supervised; batch; model-based; symbolist). The algorithm reduces the model size by learning a sparse, single shallow tree. This tree

has nodes that improve prediction accuracy and can make non-linear predictions. The final prediction is the sum of all the predictions each node provides.

SeeDot [95] is a domain-specific language (DSL) created to overcome the problem of resource-scarce MCUs not having a floating-point unit (FPU). Since most of the ML algorithms rely on the use of doubles and floats, making calculations without an FPU is accomplished by simulating IEEE-754 floating-point through software. This simulation can cause overhead and accuracy loss when making calculations; this can be deadly for any ML algorithm, especially on models exported from a different platform. The goal of SeeDot is to turn those floating-points into fixed-points, without any accuracy loss. In Section 3.4 it is shown how the use of this DSL has helped to improve the performance and accuracy of ML algorithms in resource-scarce MCUs.

CMSIS-NN [96] is a software library for neural networks (nature: supervised/reinforcement; online; model-based; connectionist) developed for Cortex-M processor cores. The software is provided by Keil [97]. Neural networks generated by the use of this tool can achieve about $4\times$ improvements in performance and energy efficiency. Furthermore, it minimizes the neural network's memory footprint. Some of the optimizations made to make the implementation possible are: fixed-point quantization; improvement in data transformation (converting 8-bits to 16-bits data type); the re-use of data, and reduction on the number of load instructions, during matrix multiplication.

FastGRNN and FastRNN [98] are algorithms to implement recurrent neural networks (RNNs) [99,100], and gated RNNs [100,101] into tiny IoT devices, and to mitigate must of the challenges they face. The use of this FastGRNN can reduce some real-world application models to fit scarce-resource MCUs. Some models are shrunk so much that can fit into MCUs with 2 Kbytes of RAM and 32 Kbytes of flash memory. The reduction of these models is achieved by adding residual connections, and the use of low-rank, sparse, and quantized (LSQ) Matrices.

TensorFlow Lite [102] has available some implementations that support a few resource-scarce platforms. The API is provided in C++, and they state it can run in a 16 KB Cortex-M3. However, as the time of writing, this tool is still experimental.

## 3.3. Field-Programmable Gate Arrays (FPGAs)

Field-programmable gate arrays (FPGAs) are powerful MCUs where the hardware can be programmed by the use of hardware description language (HDL); this is accomplished because FPGAs have several programmable logic blocks and there are connections that can be programmed to link these blocks. Besides the fact we can setup an FPGA to have high-computation powers, the authors have decided to include them in the scope of this document, because they believe, in the near future, tinier and cheaper low-power FPGAs will appear. With such availability, the window of opportunity to create end-devices by the use this technology will come along. The use of FPGA can create more dynamic, scalable, and flexible systems, that will be able to reshape as the target application needs to change. However, the authors considered target applications where the use of FPGA still meets the size and power-consumption required, and the MCU's architecture is entirely implemented in the FPGA system-on-chip (SoC).

Gopinath et al. have accelerated their DSL SeeDot using FPGA. They have compared the results of their SeeDot accelerated hardware implementation, against a typical high-level synthesis (HLS) implementation of their SeeDot C code. The authors have compared the performance against the Arduino Uno performance obtained. The HLS has scored 8x best performance in almost all the datasets, while the SeeDot hardware acceleration has achieved 128x better in almost all the datasets [95].

Kosuge et al. have used an FPGA implementation to shorten the time the iterative closest point (ICP), for object-pose estimation, took to run. 90% of the running time of the ICP was spent in the kNN search, therefore they have accelerated this search by applying a hierarchical graph instead of the conventional K-D tree. They also have reduced the time that took to generate the distances' graph by implementing a sorting-network circuit. They have been able to achieve performances of 0.6 s, while the same ICP would take 2.9 s in an SoC-GPU [103].

Irick et al. accelerated a Gaussian radial basis SVM using FPGA technology. They have been able to implement a model with 88% accuracy, which could classify 1100 images (30 × 30) in one second. They achieved this performance by analyzing the element-wise difference when calculating the Euclidean norm, which allowed them to create a 256-entry lookup table [104].

### 3.4. Resource-Scarce MCUs

The following section will focus on presenting ML models that were implemented into resource-scarce MCUs. Resource-scarce MCUs are ICPs that may have low-processing capabilities, low-frequency clocks, low-memory storage, small caching systems, and/or do not have any significant number of resources to help handle complex mathematical operations. These MCUs are generally used to perform simple control/monitoring tasks, and to focus on a single well-defined goal.

Gupta et al. have proposed the ProtoNN algorithm for implementation in resource-scarce devices (see Section 3.2). The novel algorithm was tested against other ML models, such as Neural Networks, kNN, SVM, and its accuracy was almost the same as the best model trained for each dataset. They have implemented a model for the following datasets: character recognition, MNIST, USPS, and WARD. The authors were able to implement these models into an Arduino UNO, with less than 2 kB RAM. The algorithm was able to have high performance (in terms of time), with good accuracy, and low power-consumption [93].

Kumar et al. described in their paper another novel based algorithm: Bonsai. This novel tree-based algorithm can be trained in a computer, or Cloud, and then be exported to one of the following boards: Arduino UNO; BBC Micro:Bit; or the ARM Cortex-M0. The algorithm is intended to change the paradigm of Cloud-centered systems. The accuracy is good and can be up to 30% higher than state-of-the-art ML algorithms. The goal is achieved by developing the model on a shallow, single, sparse tree learned in a low-dimensional space [94].

De Almeida et al. have implemented a multilayer perceptron (MLP) into an Arduino UNO. The system was able to identify intrusions. They have accomplished it by reducing the number of features necessary, by the use of feature extraction techniques. The neural network was made of 26 input neurons, 9 hidden layers, and 2 output neurons. The paper does not specify were the training phase had occurred [105].

Szydlo et al. have proposed a way to implement distinct ML models into an Arduino Mega, and an ARM STM32F429. They have succeeded in training the model using Python (in a personal computer), and exported the model to these MCUs. They were able to implement naive Bayes, multi-layer perceptron, and decision tree classifiers. The classifiers were able to classify images (MNIST Dataset) and to classify iris flowers (Iris Dataset). To be able to fit the models into the MCUs memory, the authors have used compiler optimizations, such as the ones provided by GNU [106].

Leech et al. developed a Bayesian model, specifically an infinite hidden markov model, that could estimate the room occupancy. The system was simple, composed by an MCU and a PIR sensor. They tested the iHMM in two ARM microcontrollers: a Cortex-M0, a low-power, low-processing microcontroller, and a Cortex-M4, with and without a floating-point unit. The authors have accomplished the goal. However, they affirm it was not possible to meet the real-time constrain without high-power consumption. They had to make several optimizations to reduce the memory footprint. The model was built in a computer, using MATLAB, and exported to C/C++. They were able to store the model into 10.36 KB of SRAM, and minimum execution time of 1.15 s, if they were using a Cortex-M4 with floating-point unit, they can have an accuracy of 80% [107].

Gobieski et al. have implemented a DNN inference model into a battery-less end-device. Their approach was bold. The principal goal is to reduce communication costs, which are highly expensive in terms of energy, making it is necessary to know when to send the sensor data to the Cloud. The only approach possible is to have a model able to make that decision on the end-device's MCU. However, battery-less devices can only perform 100,000 operations before they have to stop and recharge, which will interrupt the inference's execution. Therefore, it is necessary to ensure

the inference will continue to run after the device is recharged. The authors have accomplished this by creating three tools: GENESIS, SONIC, and TAILS. These tools are intended to compress the DNN, accelerate it, and ensure the inference can resume its running after being stopped. They have been able to implement three inferences models with these tools. An image inference system for wildlife monitoring, where the model was capable to decide if the image was from a species or not, and send it to the Cloud if it was. A human activity recognition (HAR) model in a wearable system and an audio application to recognize words in audio snippets. They also implemented an SVM but it underperformed the DNN. They have achieved accuracies above 80% in a TI-MSP430FR5994 board [108].

Haigh et al. have studied the impact each optimization they have made had, for the ML model to be able to meet the real-time and memory constraints. They have implemented an SVM, for decision making in mobile ad hoc networks (MANETs), in an ARMv7 and a PPC440 MCUs. They have chosen to optimize ML libraries available for C++. They have eliminated every package that relied on *malloc* calls, due to the time virtual memory allocation takes. To select the package to make the optimizations, they have run benchmark tests using known datasets. They have chosen to use *Weka* [63]. The authors have included multiple optimizations in terms of numerical representations, algorithmic constructs, data structures, and compiler tricks. They have profiled the calls made to the package, to see which functions were called more often. Then, they applied *inlining* to those functions and collapsed the object hierarchy to reduce the stack call. This optimization has reduced the runtime by 20%, reducing the overhead of function-call from 1 billion to 20 million. They also have performed the tests by setting the variable's type to integer 32-bits, float 32-bits, and the default 64-bits double. To achieve an integer representation, they have scaled the data by a factor F. Furthermore, they have mixed the variables' types. As expected, the double representation was slower, and the gains in accuracy were not massive enough to use them. The mixed approach (float + integer) was the fastest. However, the accuracy lost depending on the dataset, therefore this approach could not always be used. Furthermore, they simplify the kernel equations, to avoid calls to the *pow* and *sqrt* functions. However, removing these functions not always reduce the runtime, or has improved the accuracy. Such as the mixed approach, this one also dataset dependent [109].

In a different approach, Parker et al. have implemented a neural network in distinct Arduino Pro Mini (APM) MCUs. Each APM was a neuron of the neural network. This distributed neural network could learn by backpropagation and was able to dynamically learn logical operations such as XOR, AND, OR, and XNOR. The main idea was to bring true parallelism to the neural network since each chip would be responsible to hold a single neuron. The proof-of-concept was done by using four APMs, one for the input layer, two neurons in the hidden layer, and an output neuron. This neural network took 43 s to learn XOR operation, and 2 min and 46 s to learn the AND logical operation. The MCUs were connected through an I2C communication protocol [110].

Gopinath et al. have tested their SeeDot into Arduino Uno and Arduino MKR1000. The SeeDot was used to improve performance into these MCUs since it helps to generate fixed-point code, instead of floating-points, that are usually not supported by these devices. The authors have used the SeeDot DSL to improve the performance of Bonsai and ProtoNN models. The use of this optimization technique has improved the performance by at least 3x in Arduino Uno, and by at least 5x in Arduino MKR1000, except for rare occasions [95].

Suresh et al. [111] have used an kNN model for activity classification, that has achieved 96% accuracy. They have implemented the model into a Cortex-M0 processor, which helped reduce the data bytes to be sent from the end-device to the top network's layers, on a LoRA network. They have optimized the model by representing each class by s single point (centroid).

Pardo et al. implemented an ANN, in TI CC1110F32 MCU, to forecast the indoor temperature. The ANN is trained as the new data comes in from the sensors; therefore, an online learning approach was taken. The authors have created small ANN (MLP and Linear Perceptron) that were small and could fit in the MCU's memory; this was possible due to the simplicity of the problem. Because the

output is regression type, and not classification, the authors have used Measurements of Average Error (MAE) and have compared the models with a control Bayesian Baseline model. The tests were run against two datasets, one available online (Simulation), and the other obtained by the authors. In the simulation test, the results were close to the control model. The other test had shown more disparity on the results, probably because of the small number of samples obtained until the test was finished [112].

*3.5. Conclusions*

Table 2 synthesizes the information about the models' nature, discussed in the scope of this document. The authors would like to state that some algorithms can have more than one nature. Therefore, in some cases, we have presented the possible natures the algorithm can have, for further references and guidance. For more information about the algorithms nature topic, please revisit Section 2.2.1.

**Table 2.** Algorithms and their nature.

| Algorithm | Learning | Learning-Dynamics | Data Representation | Tribe |
|---|---|---|---|---|
| kNN | Supervised Unsupervised Semi-Supervised | Batch | Instance-Based | Analogizers |
| Decision Tree | Supervised | Batch | Model-Based | Symbolists |
| MLP | Supervised | Online | Model-Based | Connectionists |
| Naive Bayes | Supervised | Batch | Model-Based | Bayesians |
| RNN | Supervised | Online | Model-Based | Connectionists |
| iHMM | Supervised | Batch | Model-Based | Bayesians |
| ANN | Supervised Reinforcement | Online | Model-Based | Connectionists |
| SVM | Supervised | Batch | Model-Based Instance-Based | Analogizers |

## 4. ML in Embedded Systems: Challenges

After analyzing all the projects, novel algorithms, and tools; it is possible to perceive which are the biggest concerns when implementing an ML model into a scarce-resource MCU. The following section enumerates the concerns, and make a brief explanation on them, and how they influence the model's performance and accuracy.

*4.1. Memory Footprint*

The reduced memory storage capacity is one of the biggest challenges when implementing ML in resource-scarce MCUs. Most models will have to store some data representations as thresholds, hyperplanes, and data points; these data representations take large amounts of memory space. Therefore, the developer has to ensure it can reduce the memory footprint, without losses in accuracy.

It is important to notice that there are distinct types of memory in modern MCUs. They have Flash, SRAM, DRAM, NVRAM, and EEPROM memories (we only consider these memories because they are the ones that can be writen/re-writen more than once). All these memories have pros and cons, but perhaps the most important one is how much time it takes to access/write the data in them. The fastest are the SRAM and NVRAM; however, they are expensive; therefore, to make the MCU affordable, these memories are kept in small sizes.

Furthermore, we have to take into account that some of these memories are volatile, therefore, they can only keep the data as long as they are being powered. Therefore, data will be lost after a reboot. We must ensure, all the auxiliary calculations can be kept into a memory with fast read/write cycles, such as SRAM; that all the necessary data representations can be fit into the non-volatile memories of our systems, and that there is still space available to put our code.

Moreover, we have to also look into the number of bits each memory cell can hold. The importance of this will be explained in Section 5.6.

*4.2. Execution Time*

Time is a constraint in almost any system, being it real-time or not. It does not matter how good a piece of code is if it cannot execute in an amount of time compatible with the application. Therefore,

the first thing to ensure in any system is that the deadline is met. One of the constraints that make the algorithm run faster is the clock frequency. The *clock cycle* is what makes the hardware tick, therefore it defines how fast an processor can perform a task. However, it is not linear, we cannot assume how much time some algorithm will take to complete by using the clock frequency only. We have to also know how many *clock cycles* a *machine cycle* takes.

A *machine cycle* is the time it takes to the processor to complete these four steps: Fetch, Decode, Execute and Store. The number of *clock cycles* to finish a *machine cycle* varies from MCU to MCU, and from instruction to instruction. Therefore, in order to reduce the execution time of our algorithm we must try to make it run in as few machine cycles as possible to accomplish the task in hand.

Some arithmetic operations take longer than others. For example, multiplication can take almost three times longer to run than a sum operation, and divisions usually can take even longer. Therefore, the mathematical operations executed during the model's inference have a critical influence in execution time. More than only focusing on the arithmetic operations, we must also see the number of branch instructions, data transfer operations, and virtual data allocations that can be responsible for increasing the execution time. Especially, virtual data allocations have high overheads. However, to better understand where most of the execution time resides, it is necessary to analyze the assembly code and see the MCU's instruction set. Code analysis is a way to ensure our code is well written, and that we are only using the minimum amount of instructions needed.

### 4.3. Power Consumption

Low power consumption is a beneficial goal to be set for any project and application, and a constraint for any battery-powered device [13]. There are multiple variables which influence the power drained by our device (i.e., the hardware attached; the clock-frequency; the memory type). However, in wireless devices, 70% of its power consumption is spent in a single task: communication. Therefore, in wireless devices, the best way to reach low-power consumption is by reducing the number of communications made. In a cloud-based/fog-based system, since the devices depend on sharing data with other devices, the cost of communication is high.

The ML model implementation in the end-device allows reducing the system's communication costs. If the end-device is where the model's inference runs, the end-device no longer has to send the data to the upper layers and wait for a response. Furthermore, the deployment of an ML model to predict the data's value can ensure that the end-device only sends important data.

However, code analysis is also essential to reduce power consumption. Unnecessary calls, calculations, and complexity make the device spend more time in an awake state. Modern MCUs have low-power and/or sleep states, which reduce the power consumption to a minimum. Therefore, by ensuring that our end-device spends more time in these states, than awake, reduces the power drained by the MCU.

Another critical point is the memory type. Some memory types consume more energy than another because they may have to be refreshed from time to time. Moreover, for the same memory model, power consumption tends to be higher as its size increases.

### 4.4. Accuracy

Data is the most critical piece of a machine learning model [45]. The data used to build the model ensures how well it performs on the given problem. To perform well, the model must have data representing each case. Too few data and the algorithm cannot make any assumption, too much data and the algorithm may generalize to a useless degree. Therefore, the right amount of data is essential, and the algorithm should see the most variety of data possible to infer every possible case scenario.

The end-device uses a set of sensors to collect environmental data and then feed the new data to the built model. However, different sensors have distinct noise levels, and even different environments can account with different random noises. Therefore, when building a model, we must ensure that

the dataset used was obtained from a source similar to the one to be used to collect data for the inference phase.

This rule also applies to the learning phase. Before making the algorithm learn from a dataset, we must check the instances, features, and outputs. Some data points may have noise levels impossible to happen in real life, but that will make our model lose accuracy, or even make the model overfit the data.

### 4.5. Health

A system must be healthy to ensure that it works all the time correctly. Lack of security, privacy, and error management harm the system's health and can be deadly. Most of the times, the developers deploy the ML model and may not care about these three essential metrics. However, an ML model is a piece of software. Therefore it inherits all the software problems any other piece of code has.

An ML model has a limited number of inputs, and whenever making the inference, the data array cannot have more or fewer input values than the ones expected. Moreover, the inputs' data types cannot be changed. If the system expects a floating-point, it must not receive an integer. Therefore, if our data gets faulty, or the data acquisition system works poorly, that may break our model inference and system. Catching this kind of errors is significant, and deciding what to do when they happen is even more crucial. Otherwise, our system may find itself in a dangerous scenario.

Another metric to be careful about is our model's security or the influence the model's inference can have in the system's security. The inference triggers a response from the system. Depending on the response type, by hacking our model (i.e., by applying noise to our environment), our system can trigger an unwanted response and create many problems.

Furthermore, security and privacy come almost always hand-in-hand. The end-devices can be monitoring public places, our homes, bedrooms, and even our bodies. Therefore, they can collect personal data. Even if the data collected is used by the end-device to make the inference on the local device, and then it only sends the inference's output to the Cloud/edge; if the inference does not go somehow encrypted, anyone that accesses the inference output can use it the wrong way. An end-device like a smartwatch that can have personal data such as our heartbeat, and predict our humor from it, if hacked, can tell the attacker a lot about us.

The implementation of measures to ensure the model's security, error handling, and inference's privacy increase execution time, take memory and consume power. However, they are not optional.

### 4.6. Scalability and Flexibility

A problem that is possible to be solved by an algorithm means that by creating a piece of code that follows the algorithm's recipe, the code will solve the same problem in any platform. However, in ML, a model that is created with data from a source may not be usable in other platforms, where data can be acquired differently. Furthermore, platforms may handle data differently because of the lack of resources, which can make the models obsolete.

Flexible models are important because they ensure that their usage is not restricted to a single set of platforms or data. Moreover, flexible models can make the system's scalable, once they allow the system to be updated and to use different hardware whenever needed. The use of techniques to standardize the data and preprocess can provide more robust models. Moreover, since environment and constraints change, to know how far can our model/system scale in terms of inputs, outputs, and devices is crucial to make the system live longer.

Therefore, making models flexible and scalable may not be an easy task, but is worth pursuing. These models can ensure that models can be used in more platforms, environments, hardware, and for a more extended time. The sustainability of resources is essential because of all the monetary and time costs associated with it.

*4.7. Conclusions*

The challenges mentioned above are critical and transversal to any project which used resource-scarce MCUs in its core. However, the main focus was on how those metrics influence our model, and our model influence those metrics. The authors have decided to group the challenges into six major classes: performance, memory, power-consumption, accuracy, health, and scalability and flexibility.

Figure 6 depicts the major challenges the deployment of implementing ML algorithm in a resource-scarce MCU faces. These challenges are most times related. Therefore, by improving one of them, we can make the other worse. For example, to reduce power consumption, reducing the clock frequency is an option. However, reducing the clock frequency means the execution time increases.

Therefore, the rule is a tradeoff. The developers must look into the entire system and see what can and must be improved, and what can be sacrificed. The system's health should be the last thing to be sacrificed, and the first to be improved. The well being of our model and system is what ensures the mitigation of dangerous scenarios. The degree of importance the overthrown of a challenge has in our system is dependent on the constraints set.

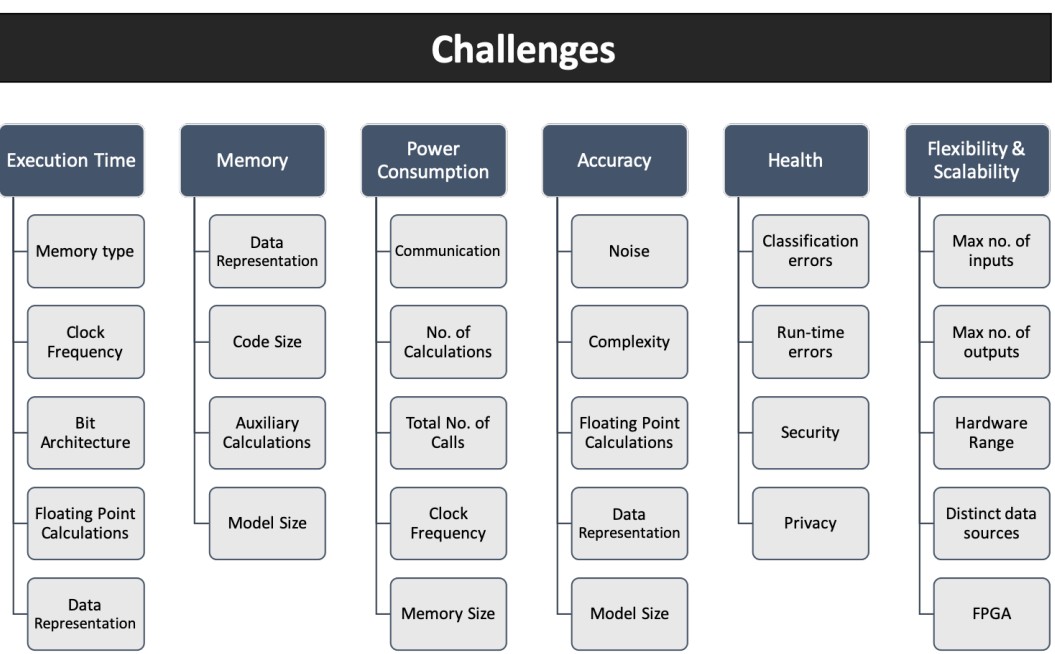

**Figure 6.** ML in embedded systems major challenges. The challenges can be grouped into six major groups: execution time; memory; power consumption; accuracy; health; flexibility and scalability. Sometimes, to improve the performance in one of the groups, another will have its perform reduced. Therefore, the developer has to do a tradeoff.

## 5. ML in Embedded Systems: Optimization Techniques

As discussed earlier, the challenges of implementing an ML model into a resource-scarce MCU are many, and most of the times a tradeoff is needed to achieve the primary goal. The following section makes an overview of the optimization techniques seen in the literature review, but also some optimizations the authors believe should be taken into account, and are not yet documented, or used.

*5.1. Test Until It Breaks*

Testing is the only fundamental tool that allows any developer to check how well its system works. Without testing a model, a piece of code, or a system, we cannot make any assumption if they

are capable of achieving their primary goal. Testing can provide insights on which are the tradeoffs that can be made.

Therefore, it is critical to test the entire system, to identify which are the data instances the system has more trouble handling. If the system can achieve the deadline much earlier than expected, but the power consumption is high, a tradeoff between performance and power consumption (e.g., by reducing the clock frequency) can be made.

### 5.2. Minimalist

When setting the goal of an ML model, the question that arises is: What features (inputs) should I use? The features define how well the model performs in the given problem. The instinct can tell us that using all the available features is the way to go. More data means our model performs better; this is not entirely true. More features can mean more complexity, and more complexity can make our system perform worse, and fall into a phenomenon called overfitting.

Therefore, before and after building and testing a model, it is essential to do something called feature selection. There are multiple techniques to perform our feature selection [113,114]. These techniques provide insights into the importance and contribution each feature has for the overall model's accuracy. However, sometimes features with a small contribution can be tie-breakers, meaning that they help in the gray areas of our data.

In conclusion, the developers should ensure to keep the model minimal, only retaining what is essential [13]. By doing so, the complexity decreases, and the memory footprint may be reduced as well. Other metrics can also be improved by reducing the model's complexity.

### 5.3. Goal-Oriented

At the beginning of each project, a goal is set. There are many ways we can develop our system to achieve the goal. In a model, the goal is the output obtained, and the correctness of the output's inference. As discussed in the previous Section 5.2, choosing the correct number of inputs is essential, but choosing the correct number of outputs can be as well, mainly when we have limited resources and time. By choosing the minimum necessary number of outputs, the complexity of our system decreases.

For example, in a classification problem, we may decide to classify between four classes (A, B, C, and D). Our system triggers a response for each classification, but we may see that a similar response is triggered for class A and B; therefore, maybe we can merge them into a single class. Alternatively, we may have the same classification problem, but the main goal is that our system never fails to distinguish A from the other classes, in this case, we can opt to reduce our model to a binary classifier.

Reducing the number of outputs does not necessarily mean that our system does not achieve the goal, for which it was designed. Therefore, whenever possible, reducing the number of outputs can bring benefits and help us overthrown some of the challenges discussed earlier.

### 5.4. Compression

Model compression can be used to reduce the memory space occupied by the model; this method can help to implement more complex ML models into such resource-scarce devices or to implement more than one model [115]. However, compressing and decompressing takes time, which must be accounted for real-time applications.

### 5.5. Platform

One main advantage of producing something to a specific MCU architecture is that we can take advantage of the resources available on the platform. When programming to a Cloud/personal computer, or anything running a standard OS, this is not so linear, because we want our code to fit many distinct platforms. When our code has to fit many distinct platforms, optimizations can only be done at the software level. However, when writing code to a specific platform, we can make optimizations at the hardware level as well. There are hardware modules specifically designed to

improve the MCU's time performance, throughput, and health. By implementing pieces of code that take the total advantages of these hardware modules, the system can improve its performance, and give us other solution that a generic piece of code would not.

Therefore, we can check if our MCU has direct memory access (DMA); which reduces the time to do memory operations [116]. Moreover, a Digital Signal Processing (DSP) unit; which may help reduce the time to make calculations, or to pre-process our data. A GPU improves performance since it can do multiple calculations simultaneously; therefore, a GPU on SoC can be useful [7]. Furthermore, an FPU is essential to reduce the time to make calculations that use floating-point numbers, and to improve accuracy.

## 5.6. Bit Architecture

To define the Computer's Architecture, the correct measurement to use is bits. A 32-bit and a 64-bit architecture have many differences, and the time to make calculations and data operations is different from one architecture to another. Therefore, to achieve real-time, the developer must also take into account the MCU's bit architecture.

A 32-bit architecture means that the datapath is 32-bit long. Therefore, each register, arithmetic logic units (ALUs), address buses, and data buses can hold only 32 bits of data at a time. Any data variable that needs more than 32 bits to be represented is split, and any data operation regarding that variable happens in multiple steps. The necessity to split the data, and make more than one operation increases the overhead. However, the same is valid for operations between data variables represented by less than 32 bits, because the system has to cast the data to fit the 32 bit architecture [117].

Therefore, to have a better performance in terms of time execution, the data variables used by our model should have the same length as our platform's architecture. But, we also have to ensure that in doing so, the model will not lose accuracy and precision.

## 5.7. Floating-Point Calculations

Most of the ML algorithms create data representations that use floating-point numbers at their core. To make floating-point calculations, most MCUs use software to simulate the IEEE-754 floating-point. Therefore, any time a floating-point operation needs to be carried, and the MCU does not support it; the CPU is responsible for dividing the operation into simpler floating-point operations. However, if the CPU does not have an FPU in it, the floating-point arithmetics are converted into fixed-point arithmetics. These conversions account in account in calculation's precision losses, time-consuming, and rounding errors [118].

An FPU can mitigate some of the errors associated with the floating-point arithmetics because this unit can perform those complex operations. However, some FPUs are only able to perform simple floating-point arithmetics, such as sum, multiplication, division, and subtraction. Therefore, even with an FPU, any complex operation (i.e., square root) not implemented into the unit, is divided into simple ones.

In conclusion, an FPU can be a beneficial resource if our model depends on floating-point data variables. However, we must ensure that our FPU can perform all the arithmetic operations necessary to reduce the overhead of splitting those complex operations in simpler ones.

## 5.8. Environment

As stated earlier, noise in the data influences the ML accuracy. Therefore, making the algorithm learn with data from distinct sources, environments, and noise levels is essential to make the model more robust. All of this is essential when the developer has no clue about the environment they will be inserted into.

However, if there is any a priori knowledge about the environment, we can reduce our model complexity to be robust only for that specific environment; this is one way of keeping the essential in

our model. Moreover, this can help in feature selection, because in some environments, features can be more correlated with the target output than others.

### 5.9. Know the Algorithm

The scientific bases for each ML algorithm's nature are distinct from algorithm to algorithm. Furthermore, the same algorithm creates a unique data representation for the given problem. Therefore, some optimizations are not transversal from algorithm to algorithm or model to model.

ML tools have created an abstraction layer, which has simplified and accelerated the process of building and testing ML models. However, after generating and testing multiple models and selecting the one to use, the best approach is to study the algorithm's bases and the model itself. By doing so, we can know which optimizations to do. For example, if the data representation generated by our algorithm falls into 9 bits, we can create unique structures to store the data in 9 bits, saving memory space. We can also design functions to handle multiple pieces of data at once during the inference phase.

One of the examples of manipulating data after the model building is an ANN. ANN should learn with high precision floating point data, but in the inference phase, the data can have lower precision [119,120]. However, this optimization may not be performed in other algorithms. Therefore, studying the algorithm and model is the only process to understand which hardware modules we can take advantage of, and which pieces of code can be re-written to improve the overall performance.

### 5.10. Cloud

The Cloud is a powerful tool; therefore, the use of Cloud computing to build a model is a way to shorten development time and to allow the developer to take an in-depth look at all the options. However, building models in the Cloud is usually done with high-level languages, which resource-scarce MCUs are not able to run and cause much overhead. In order to implement Cloud-generated models into embedded systems, new and better exporters/transpilers are needed (see Section 2.4).

The exporters must be intelligent. Because of so many platforms, optimization techniques and applications, the exporters must be able to choose and look into the problem ahead, understand it, and then chose the right code to generate. Implementing ML models capable of looking at data, and interpret it; understand the platform limitations; environment variables; measurements, and the system's goal, can create even more powerful models at lower costs.

### 5.11. Model Mesh

Complex environments can make it impossible to create minimalistic models. Therefore, we cannot reduce the number of inputs, or outputs, and the model size may be too large to fit our memory constraints. One solution is to implement the model in the edge-device or the Cloud. However, if our end-devices are in a mesh network, another approach possible is to make each node be responsible for providing an inference result, and have a central node responsible for receiving all the inference results, and deduce the output from them.

For example, if we have to distinguish between four classes (A, B, C, and D). We can make each node have a binary classifier for each class (i.e., A against all others). Therefore, each node provides true or false value for a given class. By making a node responsible for merging all the results, and make an inference from those results, we can deduce the correct class, and reduce our model size.

Meshing our model can be beneficial in situations where the connection to the Cloud is not possible, or where the edge device is not powerful enough to hold the model. Figure 7 depicts how model mesh would work.

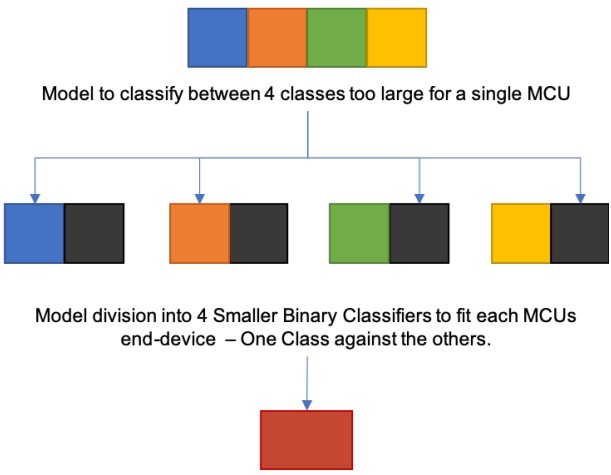

**Figure 7.** Model mesh.

## 5.12. Conclusions

The above section has presented the optimization spaces available to look into to improve and reduce any ML model. Figure 8 summarizes the optimization spaces and the principal techniques available in each one of those spaces. Besides the many spaces available to make optimizations, because of the system constraints, most of them may not be presented in our system. Understanding our system's optimization spaces is of critical importance in order to know which techniques are possible to use.

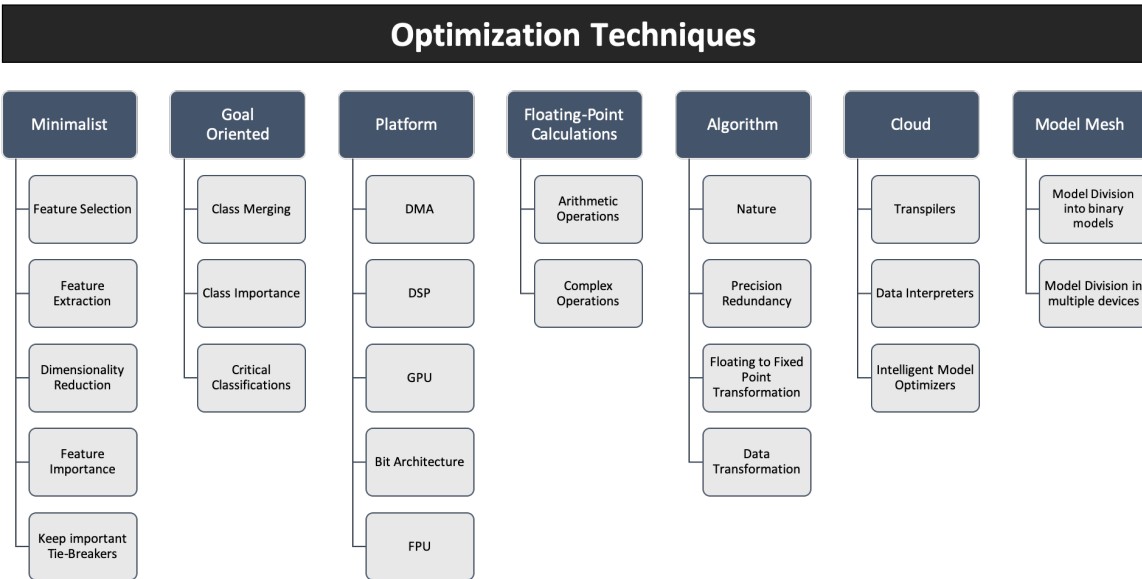

**Figure 8.** ML in embedded systems optimization techniques. There are multiple optimizations possible to make to reduce the size, improve performance, and make the system more reliable. The optimization can be done through: testing; minimalizing; goal-orientation; compression; development platform resources; bit architecture; floating-point calculations; environment; algorithm; cloud; and model meshing.

## 6. ML in Embedded Systems: Applications

There are many applications where ML models can be used to improve our system's performance. The following section centers in applications that have special significance in the end-device/resource-scarce world.

### 6.1. Communication

Probably, one of the most significant applications for ML in embedded systems is to **improve the communication** or **reduce the data traffic** between the end-device and the upper network layers. In this area of application, multiple models can be deployed. Wireless networking is not static, and the wireless signals have many variables that affect its strength, data rate, and latency (i.e., distance between nodes; propagation environment; collisions). The hardware reconfiguration can mitigate some of these issues. The reconfiguration is done through software, therefore by having a model that dynamically check our network, understands the problem, and finds the proper solution, can make communication more stable and reduce power consumption [109,121].

However, if one of the main goals is to reduce data traffic or/and database size, the model's implementation in the end-device can ensure that. If the end-device only sends the data inference result; therefore, it spends generates less data traffic, than sending the full data instance. Another way is to train our model to predict the importance of the collected data [108]. Any data scientist knows that gray areas are dangerous, and those areas usually exist due to the lack of data about them. Much of the data collected by the end-devices will be repeated. Therefore, after a while storing or sending that data can be useless. By reinforcement-learning, our end-devices can learn while collecting the data, and after a while our system can know what data is relevant, and what data is not.

### 6.2. Security and Privacy

Another key application can be to increase the system's security and privacy. The privacy and security improvements can be inherited from many ML models/algorithms. Algorithms that rely on feature extraction (i.e., ANN) create a semantics that can only be understood by the algorithm itself. Therefore, if the end-device send the data generated after the feature extraction layer, even if the data is intercepted, the attacker cannot use it [28].

### 6.3. Healthcare

Personal medical devices have several physical constraints due to the proximity to the human body; sometimes, they are even implemented inside it. Because of that, one of the most significant constraints is the device's size. To reduce the device size, the MCU must be small. Therefore, in some cases, memory storage is kept to the minimum necessary for the program to run and nothing else. Factors such as radio-frequency interference from the medical facility machines, the limits on the amount of radiation, and small batteries impose strict limits on communication [122,123].

The use of ML models in such devices can be helpful to anticipate health hazards (i.e., strokes; diabetes; cardiac arrests; seizures) and avoid casualties. However, medical devices for monitoring the human body activity have severe constraints, harsh environments, and hard-deadlines. Therefore, implementing ML models into these devices is a difficult task, but not an impossible one. It all depends on overcoming the challenges discussed earlier [124,125].

### 6.4. Industry 4.0

Industry 4.0 is an application area where ML models, data analysis, and IoT networks are beneficial and helpful tools. In production lines catching errors as soon as possible is critical, and automated ways to correct those errors is crucial to avoid production dead-times. Moreover, collecting data inside factories raise security and privacy concerns to the maximum; therefore, sending data outside the local network may not be a feasible approach [126,127].

Currently, fog-computing implementations are the most common in the factory lines. Inside the factories, the edge-device is responsible for the inference process, and to decide the actions to be done. The data does not leave the local network, but the machines, monitoring systems, and intelligent systems rely on a central system. Therefore, communication latency is still a problem, and centralizing the intelligence can make the system collapse if the central system has a hazard.

### 6.5. Autonomous Driving

Autonomous driving is probably one field of application where most of the constraints mentioned in the previous sections are presented. Most cars currently have multiple MCUs to check for problems, help during the driving process, and to warn and solve dangerous situations (e.g., stop the car before it crashes into something). Therefore, cars are complex systems where multiple components have to exchange information and interact with each other [128,129].

Due to the complexity of the car's system, the number of constraints is enormous. Each MCU has to perform its task in the time window available and communicate its results with the central computer, which receives data from all the sources and makes the final decision. The use of smart sensors, with incorporated ML models in them to provide the final classification and dedicated MCUs to make the data fusion of distinct sensors, can reduce the processing time in the central computer.

Furthermore, self-driving cars rely on technologies such as cameras, radars, and LiDARs. The quantity of data collected by those technologies is enormous, and the ML models used to make predictions are complex. With 5G, analysts predict that the cars can create a mesh network where they can share data between them, helping understand what is ahead. However, 5G dead-zones coverage can make self-driving cars to go blind in those zones [130–132]. Furthermore, with the increasing number of self-driving cars, the network's traffic can rise too much and jam. The implementation of ML models to fit in tiny MCUs and complex models capable of dealing with such amounts of data collected by the car, without relying on any communication, is an important milestone to have safe and robust self-driving cars.

### 6.6. Conclusions

There are many other fields where additional constraints exist. Because including all of them would turn the document too long, the authors decided to provide small insights about them before reaching the section's conclusions. The fields of environmental and agricultural monitoring are areas of application for ML model implementation in the end-devices. Because of the harsh conditions these devices have to endure, the size, materials, and energy consumption matter most. Furthermore, because of the remote locations, where most of them are placed, communication is not possible or too unstable to be reliable. The area those devices cover is generally big, which means that multiple devices have to be spread across it. To make such systems economically viable to be deployed, the devices have to have reduced costs.

Another field generally forgotten is underwater surveillance. The use of intelligent systems to study the seas is essential, to find new species, study environmental changes, currents, and monitor the water quality. One of the most significant constraints in this field is that underwater wireless communication does not work [133]. Therefore, there is no Cloud, fog-computing, or mesh available, which means the entire ML model has to be implemented in a single end-device. The creation of underwater surveillance devices capable of driving themselves across the sea, make predictions, and know when coming to the surface to send the collected data can improve battery life and make those systems able to cover larger areas.

There are many areas of application that can benefit from implementing the ML layer on the end-device, therefore decentralizing the ML layer. However, there are many constraints field-domain specific, which means before deciding the optimizations, materials, and MCU to use, we need to check all constraints the field of application sets. Therefore, before starting setting goals and finding the

models to work on, we need to check which are the application-specific constraints. Understanding the application field is the first step to know what technologies and models it is possible to rely on.

## 7. Future Directions

To try to set directions, in fields where new advances can make all the previous work obsolete, is not an easy task. However, there are some points of view about the future path ML in resource-scarce MCUs should follow. The following section brings awareness to the main concepts the authors believe should be followed. Figure 9 displays the road map containing all the achievements so far in the area of ML in Embedded Systems, and the key milestones the authors believe should be achieved in the next years, in order to decentralize intelligence.

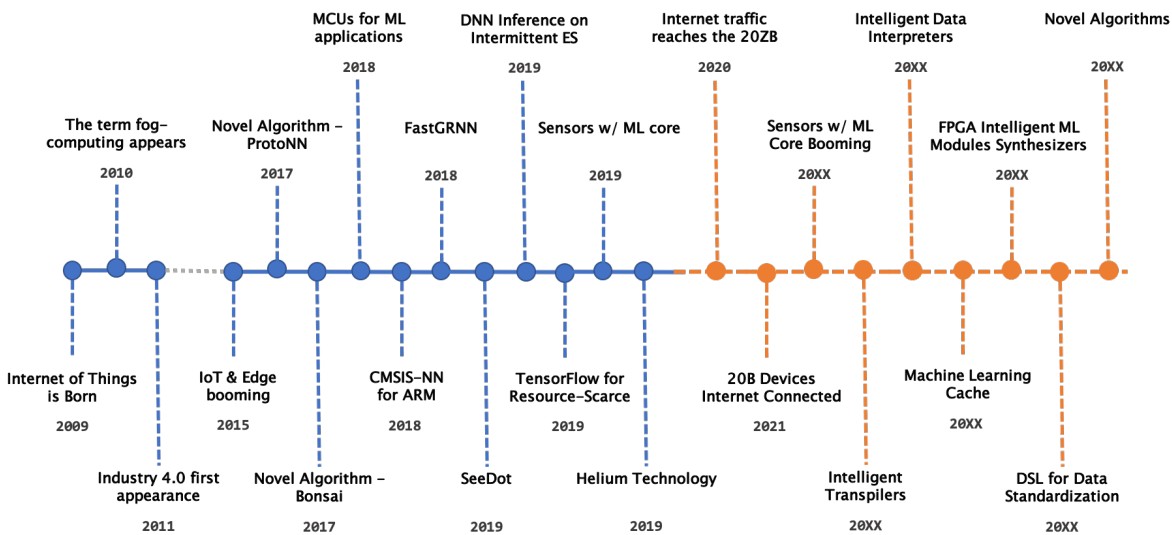

**Figure 9.** ML in embedded systems road map and future milestones.

### 7.1. Novel Algorithms

The development of new novel algorithms, based on existing ones but especially design for embedded systems, is essential. As described in Section 3, some novel algorithms can even outperform existing ones. Therefore, new advances in the field of designing ML algorithms, that take into account the restrictions most resource-scarce MCUs face, can make more and more problems to be solved by using such small devices.

### 7.2. Intelligent Transpilers

Building a model in the Cloud shorts the development time, and ensures the developer has all the resources necessary for the learning phase. Therefore, model transpilers must be designed and improved, to transpile cloud-built models to resource-scarce MCUs. However, instead of building generic transpilers that either only transpile for a single platform or language; the developer may bring intelligence to such tools, so that they learn, test new optimizations, and find the best approach to fit the requirements and constraints imposed by each system and application.

### 7.3. Intelligent Synthesizers

The same way intelligent transpilers can help to optimize code for MCUs, creating intelligent FPGA synthesizers can be helpful. Such tools can make the hardware implementation/acceleration of ML models much easier and better. Moreover, as the FPGA technology becomes cheaper and power-efficient, end-nodes capable of reshaping themselves as time and constraints change can increase the systems' flexibility and scalability.

## 7.4. Cache Machine Learning

The world is not a static place. As time advances, so does technology, environment, and society. Therefore, models may be obsolete after some periods or, from time to time, models need to be upgraded to fit new environmental constraints. Upgrading models, according to the environment or other variables, can make the models be even more minimalistic and have better performances.

There are multiple levels of cache in a computer, each one smaller than the other. In each one of those levels, data is kept according to the importance and recent usage. The use of a concept similar to cache can be applied in IoT networks. The minimalistic model is kept in the end-device, the one model to be used 90% of the time, however, since most end-devices do at least a report per day to the higher network layers; where more powerful devices exist.

The devices on the upper layers usually can have access to other sources of information (i.e., Wheater Reports). Therefore, by implementing ML learning models in these devices, which provide outputs such as new data representations, or static inputs to be used by the end-device, can be very helpful to improve the end-device overall performance. Those variables and data representation upgrades can be updated during the daily report.

## 7.5. Data

Data is the fundamental piece of the puzzle that is building an ML model. Therefore, the creation of tools to help better understand and deal with data is crucial. One of the major problems any company, data analyst, or developers face when using data from external sources is to understand how the data was collected, the units, and what it represents. Missing documentation can make a robust dataset completely obsolete and unable to be used for data fusion. The creation of a DSL to standardize the data and intelligent data interpreters that can tell which design of experiments to do, can reduce production time, and make data fusion from distinct sensors simpler.

## 8. Conclusions

The current paper has presented guidelines, taxonomies, tools, and concepts for optimization techniques to be used when implementing, and discussing ML implementation in resource-scarce MCUs, FPGAs, and end-devices. The paper shows the importance of decentralizing intelligence and bring it to the end-device. As the networks grow bigger and complex, relying on an internet connection, devices communication, and a centralized system raise hazards, concerns, and security challenges. Furthermore, internet data traffic is rising due to the data generated and exchange between IoT networks, but most of the data is useless (overhead from communication protocols), or will not be used, or even stored. Therefore, injecting such a quantity of data into the Internet only contributes to slow it down.

The implementation of ML models directly in the network's end can reduce internet traffic, mitigate latency, improve the system's security, and ensure that the system can perform in real-time. However, the challenges of implementing ML model in resource-scarce embedded systems are many and not easily overthrown. The key-concept to have in mind when making optimizations, to fit the model into the constraints imposed by our platform, is the tradeoff.

In conclusion, communication is, without any doubt, the backbone of our era. The Internet is the mainline of communication we have. However, we must ensure we have ways to survive a communication blackout; otherwise, our systems may collapse and we become blind.

**Author Contributions:** Conceptualization, S.B. and A.G.F.; methodology, S.B. and A.G.F.; writing—original draft, S.B. and A.G.F.; writing—review and editing, A.G.F. and J.C.

**Funding:** This research received no external funding.

**Acknowledgments:** This work has been supported by FCT—Fundação para a Ciência e Tecnologia within the Project Scope: UID/CEC/00319/2019.

**Conflicts of Interest:** The authors declare no conflict of interest.

**Abbreviations**

The following abbreviations are used in this manuscript:

| | |
|---|---|
| ALU | Arithmetic Logic Unit |
| ANN | Artificial Neural Networks |
| DL | Deep Learning |
| DMA | Direct Memory Access |
| DNN | Deep Neural Network |
| DSP | Digital Signal Processing |
| FastGRNN | Fast, Accurate, Stable and TinyGated Recurrent Neural Network |
| FPGA | Field-Programmable Gate Array |
| FPU | Floating-Point Unit |
| GPU | Graphical Processing Unit |
| GRNN | Gated Recurrent Neural Network [101] |
| iHMM | Infinite Hidden Markov Model |
| IoT | Internet Of Things |
| kNN | k Nearest Neighbors [134] |
| LSQ | Low-rank, Sparse, and Quantized |
| MCU | Microcontroller Unit |
| ML | Machine Learning |
| MLP | Multi-Layer Perceptron |
| NN | Neural Network |
| RNN | Recurrent Neural Network [99] |
| SoC | System-on-Chip |
| SVM | Support Vector Machine [135] |

**Appendix A. Machine Learning in Embedded Systems Overview**

Table A1 summarizes the works presented and analyzed in the scope of the document. The table contains information about the goal, hardware, model, and optimization made. If available, the table also displays information about the model's performance, memory usage, and execution time. To improve the table readability, a new row was added every time an optimization, goal, or hardware was used, even for the same project.

**Table A1.** The table summarizes the novel algorithms and ML models implemented in the resource-scarce MCUs. The table contains information about the hardware, models, and optimizations made. If available, the table shows the execution, memory, and performance obtained by the algorithm.

| Ref. | MCU | Model | Goal | Optimization | Time (s) | Size (Kb) | Acc. (%) |
|------|-----|-------|------|--------------|----------|-----------|----------|
| [93] | Arduino UNO (ATmega328P) | kNN | Character Recognition | ProtoNN [93] | 5 | 15.1 | 75.8 |
| [93] | Arduino UNO (ATmega328P) | kNN | WARD | ProtoNN [93] | 5 | 16 | 94.4 |
| [93] | Arduino UNO (ATmega328P) | kNN | MNIST | ProtoNN [93] | 5 | 16 | 93.3 |
| [93] | Arduino UNO (ATmega328P) | kNN | USPS | ProtoNN [93] | 5 | 11.6 | 94.2 |
| [94] | Arduino UNO (ATmega328P) | Decision Tree | Eye-2 | Bonsai [94] | 11–12 | 0.3–1.2 | 88 |
| [94] | Arduino UNO (ATmega328P) | Decision Tree | RTWhale-2 | Bonsai [94] | 5–7 | 0.3–1.3 | 61 |
| [94] | Arduino UNO (ATmega328P) | Decision Tree | Chars4K-2 | Bonsai [94] | 4–9 | 0.5–2 | 74 |
| [94] | Arduino UNO (ATmega328P) | Decision Tree | WARD-2 | Bonsai [94] | 5–8 | 0.5–2 | 96 |
| [94] | Arduino UNO (ATmega328P) | Decision Tree | CIFAR10-2 | Bonsai [94] | 5–8 | 0.5–2 | 73 |
| [94] | Arduino UNO (ATmega328P) | Decision Tree | USPS-2 | Bonsai [94] | 3–6 | 0.5–2 | 94 |
| [94] | Arduino UNO (ATmega328P) | Decision Tree | MNIST-2 | Bonsai [94] | 5–9 | 0.5–2 | 94 |
| [95] | Arduino UNO (ATmega328P) | ProtoNN | cifar-2 | SeeDot | 14.1 | <32 | - |
| [95] | Arduino UNO (ATmega328P) | ProtoNN | cr-2 | SeeDot | 28.4 | <32 | - |
| [95] | Arduino UNO (ATmega328P) | ProtoNN | cr-62 | SeeDot | 34.7 | <32 | - |
| [95] | Arduino UNO (ATmega328P) | ProtoNN | curet-61 | SeeDot | 37.8 | <32 | - |
| [95] | Arduino UNO (ATmega328P) | ProtoNN | letter-26 | SeeDot | 35.4 | <32 | - |

**Table A1.** *Cont.*

| Ref. | MCU | Model | Goal | Optimization | Time (s) | Size (Kb) | Acc. (%) |
|---|---|---|---|---|---|---|---|
| [95] | Arduino UNO (ATmega328P) | ProtoNN | mnist-2 | SeeDot | 16 | <32 | - |
| [95] | Arduino UNO (ATmega328P) | ProtoNN | mnist-10 | SeeDot | 18.5 | <32 | - |
| [95] | Arduino UNO (ATmega328P) | ProtoNN | usps-2 | SeeDot | 9.2 | <32 | - |
| [95] | Arduino UNO (ATmega328P) | ProtoNN | usps-10 | SeeDot | 14 | <32 | - |
| [95] | Arduino UNO (ATmega328P) | ProtoNN | ward-2 | SeeDot | 23.2 | <32 | - |
| [95] | Arduino UNO (ATmega328P) | BONSAI | cifar-2 | SeeDot | 11.8 | <32 | - |
| [95] | Arduino UNO (ATmega328P) | BONSAI | cr-2 | SeeDot | 11.9 | <32 | - |
| [95] | Arduino UNO (ATmega328P) | BONSAI | cr-62 | SeeDot | 29 | <32 | - |
| [95] | Arduino UNO (ATmega328P) | BONSAI | curet-61 | SeeDot | 39.7 | <32 | - |
| [95] | Arduino UNO (ATmega328P) | BONSAI | letter-26 | SeeDot | 11.2 | <32 | - |
| [95] | Arduino UNO (ATmega328P) | BONSAI | mnist-2 | SeeDot | 11.6 | <32 | - |
| [95] | Arduino UNO (ATmega328P) | BONSAI | mnist-10 | SeeDot | 16.8 | <32 | - |
| [95] | Arduino UNO (ATmega328P) | BONSAI | usps-2 | SeeDot | 11.6 | <32 | - |
| [95] | Arduino UNO (ATmega328P) | BONSAI | usps-10 | SeeDot | 9.2 | <32 | - |
| [95] | Arduino UNO (ATmega328P) | BONSAI | ward-2 | SeeDot | 14.3 | <32 | - |
| [105] | Arduino UNO (ATmega328P) | MLP | Intrusion Detection | Memory Footprint | 6 | - | 97.1 |
| [106] | Arduino Mega (ATmega2560) | Naive Bayes | Iris Flower Classification | Memory Footprint | - | 2.3 | 100 |
| [106] | Arduino Mega (ATmega2560) | MLP | Iris Flower Classification | Memory Footprint | - | 2.4 | 93.3 |
| [106] | Arduino Mega (ATmega2560) | Decision Tree | Iris Flower Classification | Memory Footprint | - | 0.3 | 93.3 |

**Table A1.** *Cont.*

| Ref. | MCU | Model | Goal | Optimization | Time (s) | Size (Kb) | Acc. (%) |
|------|-----|-------|------|--------------|----------|-----------|----------|
| [106] | Arduino Mega (ATmega2560) | MLP | Image Recognition | Memory Footprint | - | 52 | 91.9 |
| [106] | Arduino Mega (ATmega2560) | Decision Tree | Image Recognition | Memory Footprint | - | 76–166 | 87.4 |
| [95] | Arduino MKR1000 (Cortex-M0+) | BONSAI | cifar-2 | SeeDot | 2.8 | <32 | - |
| [95] | Arduino MKR1000 (Cortex-M0+) | BONSAI | cr-2 | SeeDot | 2.8 | <32 | - |
| [95] | Arduino MKR1000 (Cortex-M0+) | BONSAI | cr-62 | SeeDot | 5.7 | <32 | - |
| [95] | Arduino MKR1000 (Cortex-M0+) | BONSAI | curet-61 | SeeDot | 6.2 | <32 | - |
| [95] | Arduino MKR1000 (Cortex-M0+) | BONSAI | letter-26 | SeeDot | 1.8 | <32 | - |
| [95] | Arduino MKR1000 (Cortex-M0+) | Decision Tree | mnist-2 | SeeDot and Bonsai | 3 | <32 | - |
| [95] | Arduino MKR1000 (Cortex-M0+) | BONSAI | mnist-10 | SeeDot | 3.8 | <32 | - |
| [95] | Arduino MKR1000 (Cortex-M0+) | BONSAI | usps-2 | SeeDot | 2.7 | <32 | - |
| [95] | Arduino MKR1000 (Cortex-M0+) | BONSAI | usps-10 | SeeDot | 2 | <32 | - |
| [95] | Arduino MKR1000 (Cortex-M0+) | BONSAI | ward-2 | SeeDot | 3.6 | <32 | - |
| [95] | Arduino MKR1000 (Cortex-M0+) | ProtoNN | cifar-2 | SeeDot | 2.6 | <32 | - |
| [95] | Arduino MKR1000 (Cortex-M0+) | BONSAI | cr-2 | SeeDot | 4.8 | <32 | - |
| [95] | Arduino MKR1000 (Cortex-M0+) | BONSAI | cr-62 | SeeDot | 5.1 | <32 | - |
| [95] | Arduino MKR1000 (Cortex-M0+) | BONSAI | curet-61 | SeeDot | 5.6 | <32 | - |
| [95] | Arduino MKR1000 (Cortex-M0+) | BONSAI | letter-26 | SeeDot | 4.4 | <32 | - |

Table A1. *Cont.*

| Ref. | MCU | Model | Goal | Optimization | Time (s) | Size (Kb) | Acc. (%) |
|------|-----|-------|------|--------------|----------|-----------|----------|
| [95] | Arduino MKR1000 (Cortex-M0+) | BONSAI | mnist-2 | SeeDot | 3.3 | <32 | - |
| [95] | Arduino MKR1000 (Cortex-M0+) | BONSAI | mnist-10 | SeeDot | 3.7 | <32 | - |
| [95] | Arduino MKR1000 (Cortex-M0+) | BONSAI | usps-2 | SeeDot | 1.5 | <32 | - |
| [95] | Arduino MKR1000 (Cortex-M0+) | BONSAI | usps-10 | SeeDot | 2.5 | <32 | - |
| [95] | Arduino MKR1000 (Cortex-M0+) | kNN | ward-2 | ProtoNN and SeeDot | 4 | <32 | - |
| [98] | Arduino MKR1000 (Cortex-M0+) | RNN | Google-12 | Fast(G)RRN | 5–2000 | 6–56 | 92–93 |
| [98] | Arduino MKR1000 (Cortex-M0+) | RNN | HAR-2 | Fast(G)RRN | 162–553 | 6–56 | 92–93 |
| [98] | Arduino MKR1000 (Cortex-M0+) | RNN | Wakeword-2 | Fast(G)RRN | 175–755 | 1–8 | 97–98 |
| [107] | ARM Cortex-M0 | Infinite Hidden Markov | Room Occupancy | Memory Footprint | 22 | 22 | 80 |
| [111] | ARM Cortex-M0 | kNN | Activity Monitoring | Cluster reduction to a single point | - | - | 96.2 |
| [106] | ARM Cortex-M4 | Naive Bayes | Iris Flower Classification | Memory Footprint | - | 2.0–3.4 | 100 |
| [106] | ARM Cortex-M4 | MLP | Iris Flower Classification | Memory Footprint | - | 3.9–5.2 | 93.3 |
| [106] | ARM Cortex-M4 | Decision Tree | Iris Flower Classification | Memory Footprint | - | 0.02–0.6 | 93.3 |
| [106] | ARM Cortex-M4 | Naive Bayes | Image Recognition | Memory Footprint | - | 191 | 55.6 |
| [106] | ARM Cortex-M4 | MLP | Image Recognition | Memory Footprint | - | 52.3–55 | 91.9 |
| [106] | ARM Cortex-M4 | Decision Tree | Image Recognition | Memory Footprint | - | 54–159 | 87.4 |
| [107] | ARM Cortex-M4 | Infinite Hidden Markov | Room Occupancy | Memory Footprint | 9550 | 22 | 80 |
| [107] | ARM Cortex-M4 | Infinite Hidden Markov | Room Occupancy | Memory Footprint; Floating Point Unit | 1150 | 22 | 80 |
| [112] | TI CC1110F32 | ANN | Indoor Temperature Forecast | Memory Footprint | - | - | -% |

**Table A1.** *Cont.*

| Ref. | MCU | Model | Goal | Optimization | Time (s) | Size (Kb) | Acc. (%) |
|---|---|---|---|---|---|---|---|
| [108] | MSP430FR5994 w/100 uF | DNN Inference | Image Recognition | Compression using separation and pruning techniques; Intermittent Running; | 8000 | - | 99% |
| [108] | MSP430FR5994 w/100 uF | DNN Inference | Image Recognition | Compression using separation and pruning techniques; Intermittent Running; TI Low Energy Accelerator (LEA) | 5000 | - | 99% |
| [108] | MSP430FR5994 w/100 uF | DNN Inference | Human Activity Recognition | Compression using separation and pruning techniques; Intermittent Running; | 5000 | - | 88% |
| [108] | MSP430FR5994 w/100 uF | DNN Inference | Human Activity Recognition | Compression using separation and pruning techniques; Intermittent Running; TI Low Energy Accelerator (LEA) | 2000 | - | 88% |
| [108] | MSP430FR5994 w/100 uF | DNN Inference | Audio Word Recognition | Compression using separation and pruning techniques; Intermittent Running; | 15,000 | - | 84% |
| [108] | MSP430FR5994 w/100 uF | DNN Inference | Audio Word Recognition | Compression using separation and pruning techniques; Intermittent Running; TI Low Energy Accelerator (LEA) | 10,000 | - | 84% |

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
