# Peer review of "Machine Learning in Resource-Scarce Embedded Systems, FPGAs, and End-Devices: A Survey"

_electronics, doi:10.3390/electronics8111289_

Round 1
Reviewer 1 Report
The core idea of this manuscript, decentralize the network’s intelligence, is very interesting. There is many interesting and useful information in this survey. But the survey lacks a comprehensive review in the light of technical evolutions. The road map of Machine Learning in End-Device Embedded Systems with key milestones is also missing. Without these guides, the information in the manuscript is still fragmented.
Author Response
Please, see the file attached.

Reviewer 2 Report
This journal is a survey paper on machine learning techniques and considerations for machine learning with limited resource end devices. This reviewer would like to ask a few question to the authors as the following questions:
I agreed that the target of this paper is an embedded system or resource constrained end device based machine learning system. However, I wonder if FPGAs can be discussed on the same line as the previous system. Usually FPGA shows very high performance in term of computation. It is necessary to reconsider the FPGA part entering to the scope of this paper.
As time goes by, the performance of hardware is evolving, so it is necessary to consider how to define the definition criteria of end device embedded system. For example, what meaning is given to the criteria set in this paper?
Creating a table that summarizes the algorithms mentioned in Chapter 3 is likely to make it easier for readers to identify the compared algorithms.
As a major challenge, it is necessary to look at the following additional features.
scalability, flexibility, real-time
For the applications where end device machine learning is used, we need to check if there are any additional requirements.
Agriculture, autonomous driving, environmental information collection, etc.
The description in Appendix A is not clearly visible.
Please check your paper. There are some typos.
example: Line 138: section ?? / 175 line: we we
Author Response
Please, see the file attached.

Reviewer 3 Report
The authors carry out an extensive and interesting work on the state of the art in the application of machine learning techniques oriented to embedded systems with resource-scarce, FPGAs and end- devices.
The document is well justified and the authors provide the appropriate bibliographic references, where the different technological scenarios and contexts related to machine learning are described and analysed. As the authors comment, the document aims to provide taxonomies, concepts and guidelines to help decentralize the network’s intelligence in the near future.
Particularly interesting is the contribution that the authors make in tables 1 and A1, where they reflect different comparisons of the application of machine learning in different resource-scarce MCUs existing in the market.
Given the length of the article, it is understood that the authors have tried to strike a balance between the contents dealt with and the inclusion of figures and schemes. The presented work has a clear structure, however, due to the amount of concepts it handles, its reading becomes a little tired. In this sense, it would be advisable to use a few more graphs (especially in chapter 5), in order to synthesize the concepts dealt with and make it easier for the reader to remember and classify them.
The general impression of the work is very good, it has the interest and quality to be accepted.
Author Response
Please, see the file attached.

Round 2
Reviewer 1 Report
Congratulations on the completion of a high quality manuscript.